# Improved GNSS-Based Bistatic SAR Using Multi-Satellites Fusion: Analysis and Experimental Demonstration

**DOI:** 10.3390/s20247119

**Published:** 2020-12-11

**Authors:** Shiyu Wu, Dongkai Yang, Yunlong Zhu, Feng Wang

**Affiliations:** School of Electronics and Information Engineering, Beihang University, Beijing 100191, China; shiyu_wu@buaa.edu.cn (S.W.); edkyang@buaa.edu.cn (D.Y.); wangfeng_buaa@buaa.edu.cn (F.W.)

**Keywords:** GNSS, SAR, PSF, two-dimensional (2-D) resolution, multi-satellites fusion, greedy algorithm

## Abstract

The Global Navigation Satellite System (GNSS)-based Bistatic Synthetic Aperture Radar (SAR) is getting more and more attention in remote sensing for its all-weather and real-time global observation capability. Its low range resolution results from the narrow signal bandwidth limits in its development. The configuration difference caused by the illumination angle and movement direction of the different satellites makes it possible to improve resolution by multi-satellite fusion. However, this also introduces new problems with the resolution-enhancing efficiency and increased computation brought about by the fusion. In this paper, we aim at effectively improving the resolution of the multi-satellite fusion system. To this purpose, firstly, the Point Spread Function (PSF) of the multi-satellite fusion system is analyzed, and focusing on the relationship between the fusion resolution and the geometric configuration and the number of satellites. Numerical simulation results show that, compared with multi-satellite fusion, dual-satellite fusion is a combination with higher resolution enhancement efficiency. Secondly, a method for dual-satellite fusion imaging based on optimized satellite selection is proposed. With the greedy algorithm, the selection is divided into two steps: in the first step, according to geometry configuration, the single-satellite with the optimal 2-D resolution is selected as the reference satellite; in the second step, the angles between the azimuthal vector of the reference satellite and the azimuthal vector of the other satellites were calculated by the traversal method, the satellite corresponding to the intersection angle which is closest to 90° is selected as the auxiliary satellite. The fused image was obtained by non-coherent addition of the images generated by the reference satellite and the auxiliary satellite, respectively. Finally, the GPS L1 real orbit multi-target simulation and experimental validation were conducted, respectively. The simulation results show that the 2-D resolution of the images produced by our proposed method is globally optimal 15 times and suboptimal 8 times out of 24 data sets. The experimental results show that the 2-D resolution of our proposed method is optimal in the scene, and the area of the resolution unit is reduced by 70.1% compared to the single-satellite’s images. In the experiment, there are three navigation satellites for imaging, the time taken to the proposed method was 66.6% that of the traversal method. Simulations and experiments fully demonstrate the feasibility of the method.

## 1. Introduction

Apart from providing the global positioning and timing services, the Global Navigation Satellite System GNSS could be used as the transmitters in bistatic radar to explore the earth parameters and detect targets all-day and all-weather. This technology is known as GNSS-Reflectometry (GNSS-R) which has the advantage of low-cost, global coverage because of no transmitters and massive GNSS satellites. At present, GNSS-R has been used to measure sea-surface level [1,2], retrieve wind speed [3,4], estimate soil moisture and biomass [5,6,7,8], detect oil spill [9] et al. Additionally, the imaging capacity of the GNSS-R signal has been also taken into account with the development of technologies.

GNSS-based Bistatic SAR is one emerging application that combines GNSS-R and bistatic SAR imaging technology. The combination of the two technologies can be selected at the signal level, navigation satellites are used as the non-cooperative illuminator, and the bistatic SAR mode is used for echo data collection and processing to generate imaging. The feasibility of the GNSS-based Bistatic SAR has been demonstrated with GPS [10,11], GLONASS [12], Beidou [13], and Galileo [14] as illuminators. Furthermore, a series of problems encountered during the research process were studied in depth, such as power budget [15], synchronization [16], and imaging algorithms [17]. This technology has been demonstrated to be meaningful for detecting [18], recognizing targets [19], and surface change monitoring [20,21]. Moreover, the number of GNSS satellites illuminating the same area on the ground from different aspect angles is about 20–30 for GNSS constellations. This factor makes it possible to select the best bistatic geometry or combine multiple satellites for observation. For the radar itself, it can improve resolution performance and reduce shadow area. In terms of remote sensing applications, it can effectively improve the probability of detection [22] and the accuracy of surface deformation monitoring [23].

GNSS signals are not designed for radar or remote sensing, prompting the introduction of several problems in radar and remote sensing applications. First of all, the power budget issue is affected by the transmitting power of navigation satellites and the orbit height (navigation satellites are mainly medium earth satellites: MEO, Medium Earth Orbit), resulting in very low power density near the surface of the earth. Secondly, the navigation signal bandwidth is relatively narrow to the traditional synthetic aperture radar signal, the range resolution is limited to the signal bandwidth. For example, the bandwidth of the GPS L1 signal is 2.046 MHz, which provides a 3 dB range resolution about 87.9 m in an equivalent monostatic model. With the further development of the GNSS systems, it provides a wider signal bandwidth. The GPS L5, Beidou B3, GALILEO E5a and E5b signals provide a 3 dB range resolution about 8.79 m. Since GNSS-based SAR is in the category of bistatic, the geometric configuration may greatly deteriorate the spatial resolution of the system [24], Therefore, the application field of this technology is limited.

To tackle the aforementioned problem, researchers have proposed different methods to improve imaging resolution. A sufficiently high signal-to-noise ratio (SNR) can be obtained by increasing the synthetic aperture time, which can be a few minutes or even longer for a fixed receiver [25]. Due to the orbit curvature of the navigation satellite, the long dwell time can achieve a substantial increase in the spatial resolution of the system [26]. One or more receivers receive multiple navigation satellites signals to generate different SAR images of the same target area, and the images can be fused in a coherently or incoherently way to improve the spatial resolution, it has been verified the feasibility and performance by experiments [27,28,29]. A large number of satellites with different views can be used to enhance the information space of the images, the feature value can be extracted from each image, and then the feature-level combination can be used to significantly enhance the target information [30], conducive to feature extraction, feature recognition, and classification of the target area. In [31], using the B3 signal of the BeiDou navigation system to study the segmentation method of the sensing area, the method is based on the changes of the radar cross section (RCS) of different targets under different bistatic geometric configurations, and is verified by experiments. With Galileo E5 echo imaging, the entire E5 channel can be stitched together using a spectral equalization technique to increase the 3 dB range resolution to 1.758 m, but this technique is limited to alternate binary offset carrier (AltBOC) signal and introduces essential reduction in signal-noise ratio (SNR) [14].

It has been proved that the resolution of the system can be improved by using space separated multi-satellite and multi-station imaging, however, these methods introduce new problems. The first is the increased computation, and the second is the resolution improvement efficiency problem caused by multi-satellite fusion. In this paper, to solve this problem, a multi-satellite fusion method with improved imaging resolution is proposed based on an optimized satellite selection strategy. We consider a ground-based stationary receiver configuration, but without loss of generality, the method proposed can be used in others topologies. The processing rules for signals from multiple satellites are combined based on multiple parameters (the transmitted signal, the configuration of the transceiver station, and the synthetic aperture time) defined by Multi-Satellite Fusion Point Spread Function (MPSF). In this paper, we focus on the influence of transceiver station configuration on the fusion resolution.

We established an accurate geometric configuration model and evaluated the influence of geometric configuration on the fusion resolution in the form of numerical simulation. Simulation results show that multi-satellite fusion can not only improve system resolution but also potentially reduce it, which is mainly affected by the geometric configuration of multiple satellites. The resolution improvement efficiency decreases with increasing number of satellites, and the geometric configuration requirements increase as the number of satellites increases. Additionally, the simulation reveals that dual-satellite fusion is a better cost-effective option because of the increased resolution efficiency. A fusion imaging method based on optimized satellite selection is proposed to realize the effective spatial resolution improvement of a multi-satellite system. 

To demonstrate the feasibility of the proposed imaging algorithm, point target simulations were performed using real GNSS orbit data. The GPS L1 satellite is chosen as the illuminator. A total of 24 sets of data were collected in one day, and the proposed method was compared with the traditional traversal fusion method. The proposed method obtained optimal resolution images 15 times and suboptimal images 8 times, demonstrating the feasibility of the proposed algorithm.

An experiment is also conducted to further confirm the feasibility of the proposed algorithm. The GPS L1 satellite is chosen as the illuminator. The Beihang University Gymnasium was selected as the imaging target because of the large RCS. The collected echo data are processed using the traversal fusion method and the proposed algorithm, respectively. The proposed algorithm produces the optimal resolution image of the experimental scene, the time taken is 66.6% of the traversal fusion method, demonstrating the feasibility of the proposed algorithm.

The structure of this article is as follows: numerical analysis of PSF and MPSF are performed in Section 2, a fusion imaging method based on optimize satellite selection is introduced in Section 3, the simulation is carried out through the real navigation satellite orbit data, and the simulation results are analyzed in Section 4, the proposed method is verified and analyzed experimentally in Section 5, finally we summarize our conclusions in Section 6.

## 2. PSF of GNSS-Based Bistatic SAR System

### 2.1. Generalized Ambiguity Function

In bistatic SAR, the ambiguity function (AF) of bistatic radar is not only related to the transmitted signal but also related to factors such as the bistatic topology structure. According to the research of M. Cherniakov and Tao Zeng et al. [32], the AF was extended to the GNSS-based Bistatic radar, thus the generalized ambiguity function (GAF) could be expressed as:(1)$$\chi (\vec{OA},\vec{OB})=K\int s_{A}(t,u)s_{B}^{*}(t,u)dtdu$$
where $s_{A}(t,u)$ and $s_{B}(t,u)$ are the GNSS baseband echo signals by a point target with unit radar cross section and located at the ground points A and B. *O* is center of the scene, $\vec{OA}$ and $\vec{OB}$ are the vectors from point O to point A and B, respectively. The variable u is called fast time, constant during the pseudo-random code period *T* (1 ms long for the GPS L1(C/A-code)), t denote the slow-time, set t=nT+u. K is used as normalization parameter such as $\chi \left|\left(\vec{OA},\vec{OA}\right)\right|=1$. In the article [33], the GAF simplified as follows:(2)$$\chi (\vec{r})=Kg(\tau_{d}(\vec{r})M(f_{d}(\vec{r}))$$
where $\vec{r}=\vec{OA}-\vec{OB}$, g(t) is the matched filter output of the ranging signal, since navigation satellites emit continuous pseudo-random signals, the output of the matched filter is a triangular wave, which can be expressed as:(3)$$g(\tau_{d})\approx P_{0}\cdot tri(\pi B\tau_{d})$$
where *B* is the navigation signal bandwidth, $P_{0}$ is power, tri(⋅) is triangular function, specifying that the range resolution. The delay $\tau_{d}$ is related to the geometric configuration, which can be expressed as:(4)$$\tau_{d}=\frac{\vec{\beta }(u_{ta})\cdot \vec{r}}{c}$$
where $\vec{\beta }(u)$ is the bisector of the bistatic angle, c is the speed of light, $u_{ta}$ is the value of u evaluated at the midpoint of *T* ($u\in [u_{ta}-T/2,u_{ta}+T/2]$). m(u) represents the ratio of signal received power to transmit power, M(f) is the inverse Fourier transform of m(u), $M\left(f_{d}\right)$ is given by:(5)$$M\left(f_{d}\right)\approx \sin c(\pi Tf_{d})$$

sinc(⋅) specifies azimuth resolution. $f_{d}$ is Doppler frequency, which is generated in two parts: the motion of the receiver and the motion of the navigation satellite, which can be expressed as:(6)$$f_{d}=f\frac{d\tau_{d}}{du}=\frac{1}{\lambda }\frac{d\beta (u_{ta})}{du}\cdot \vec{r}$$
where f is the navigation signal carrier frequency, λ is wavelength of navigation signal.

### 2.2. PSF of Single-Satellite

#### 2.2.1. Resolution Analysis Parameter

We consider a ground-based stationary receiver collecting the signals emitted from a GNSS transmitter and reflected by a stationary scene. Figure 1 shows the geometric configuration, where the navigation satellite S is located on the OXYZ plane, the sub-satellite point $S^{\prime }$ is located on the negative semi-axis of the Y-axis, R is a ground-based stationary receiver, and $R^{\prime }$ is the ground mapping point, the satellite velocity $\vec{V_{T}}$ points to the positive direction of the X-axis, $\theta_{T_{x}}$ and $\theta_{R_{x}}$ is the ground incident angles of the navigation satellite and the receiver relative to the center *O* of the target area, the observation angle of the navigation satellite ϕ is the angle with $\vec{OR^{\prime }}$ and $\vec{OS^{\prime }}$, *P* is any point in the scene, $\vec{OP}$ is the vector from point *O* to point *P*, $\vec{OP}=\vec{r}$, $\vec{i_{Tx}}$ and $\vec{i_{Rx}}$ are the unit vectors of *P* pointing to the navigation satellite and receiver, respectively. The ground range and azimuth resolution vector in the case of bistatic are obtained by calculating the 3 dB bandwidth of the sum function as follows:(7)$$\vec{\delta_{rg}}=a_{r}\frac{c}{2Bcos\left(\frac{\beta }{2}\right)cosa}\cdot \vec{i_{rg}},\;\vec{\delta_{ag}}=a_{a}\frac{\lambda }{2\omega Tcosa}\cdot \vec{i_{ag}}$$
where $a_{r}$ is a factor accounting the shape of g(t); because the signal transmitted by the navigation satellite is a pseudo-random code, the output of the matched filter is a triangular wave, for its 3dB attenuation, $a_{r}=0.586$. $a_{a}$ is a factor accounting the shape of M(f); between the synthetic aperture time, the power of the received signal can be approximated as a rectangular function, so the output of the matched filter is a sinc function, which attenuates 3 dB, $a_{a}=0.886$. $\vec{\beta }$ is the angular bisector vector of the satellite and receiver position, ω is the equivalent bistatic angular speed. The bistatic angle can be expressed as:(8)$$\beta =cos^{-1}\langle \vec{i_{Tx}},\vec{i_{Rx}}\rangle$$

a is the angle between the range and azimuth vector direction with the ground plane.
(9)$$a=arccos\langle \vec{\beta },\vec{r}\rangle$$

Since a ground-based stationary receiver, the imaging area will be relatively small, take the center *O* of the target area as a reference to calculate the ground-to-ground resolution and azimuth resolution. The unit vectors of range and azimuth projected to the ground are respectively:(10)$$\vec{i_{ag}}=\left[\begin{array}{l} \frac{-sin\theta_{Rx}cos\phi -sin\theta_{Tx}}{\sqrt{sin^{2}\theta_{Tx}+sin^{2}\theta_{Rx}+2sin\theta_{Tx}sin\theta_{Rx}cos\phi }}\\ \frac{sin\theta_{Rx}sin\phi }{\sqrt{sin^{2}\theta_{Tx}+sin^{2}\theta_{Rx}+2sin\theta_{Tx}}sin\theta_{Rx}cos\phi } \end{array}\right]$$
(11)$$\vec{i_{rg}}=\frac{\left({\vec{V}}_{Tx}-\left({\vec{V}}_{Tx}\cdot {\vec{i}}_{Tx}\right){\vec{i}}_{Tx}\right)}{\left|\left({\vec{V}}_{Tx}-\left({\vec{V}}_{Tx}\cdot {\vec{i}}_{Tx}\right){\vec{i}}_{Tx}\right)\right|}$$

#### 2.2.2. PSF Numerical Analytical

To more intuitively evaluate the two-dimensional (2-D) capability of GNSS-based Bistatic SAR, 2-D resolution unit was proposed in [32] to analyze the resolution capability of bistatic SAR, using the 3dB approximate elliptical area of its Point Spread Function (PSF) to characterize it [33]. The (PSF) is the result of (2) being mapped onto the ground plane, it is an approximately elliptical analytic unit. The PSF function is given by:(12)PSF(r→)=∫uta−T2uta+T2∫f−B/2f+B/2exp(−j2πfcβ→(u)⋅r→)dfdu

The resolution ellipse is computed as:(13)$$\mathrm{PSF}(\vec{r})=\frac{1}{\sqrt{2}}$$

Formula (12) shows different bistatic PSF parameters are affected by the following factors: (1) synthetic aperture time *T*, (2) signal bandwidth *B*, (3) geometric configuration of bistatic. As shown in Figure 1, the influence of geometrical configuration on PSF is mainly considered, determined by navigation satellite position information ($\theta_{T_{x}}$, $\theta_{R_{x}}$ and ϕ) and the velocity $\vec{V}$ of the navigation satellite. Based on these 4 parameters we build accurate numerical simulation scenarios, assuming that the GPS L1 satellite is moving in a straight line at a constant speed. The specific parameters are shown in Table 1, and the simulation scene as shown in Figure 2, the results of PSFs can be evaluated using both numerical [33] and analytical methods, which can be expressed as $A_{1}(\varphi ,\gamma ,\alpha )$, where α is the complementary angle of $\theta_{T_{x}}$, φ is the supplementary angle of ϕ, $\theta_{R_{x}}$,γ is the angle with the satellite velocity $\vec{V_{s}}$ and the positive direction of the X-axis.

As shown in Figure 3a, the simulated optimal PSF $A_{1}(180^{\circ },90^{\circ },45^{\circ })$, η is the resolution ellipse’s orientation, which is the function of the $\vec{\delta_{rg}}$, $\vec{\delta_{ag}}$ and the angle ε between $\vec{i_{rg}}$ and $\vec{i_{rg}}$. At the time of collection, the bistatic geometric configuration is close to the monostatic mode, the range direction is nearly orthogonal to the azimuth direction ($\varepsilon =90^{\circ }$,$\eta \approx 0^{\circ }$), the area of 2-D resolution is 1390 m^2^. The azimuth resolution is mainly derived from velocity of the navigation satellite and it changes of different configurations are relatively minor in a ground-based stationary receiver. Still, it is still necessary to pay attention to the direction of azimuth vector to follow the changes with the position and velocity direction of navigation satellite. The range resolution varies with the change of the azimuth angle and pitch angle. Consequently, the change of azimuth angle φ and velocity direction γ are the main reason for the change of 2-D resolution in simulation scenario. With changes in φ and γ, resulting in changes in the PSF resolution elliptic parameters, a simulated PSF $A_{1}(150^{\circ },60^{\circ },45^{\circ })$ is shown in Figure 3b. It’s 2-D resolution area is 2397 m^2^, where $\varepsilon =39^{\circ }$ and $\eta \approx 59^{\circ }$. Figure 4 shows the distribution of 2-D resolution units area produced to follow the change of φ and γ, the highlighted part shows that the area of the 2-D resolution unit is very large, indicating that the 2-D resolution capability extremely deteriorates under this geometric configuration.

### 2.3. MPSF with Non-Coherent Summation

The multi-satellite fusion system is a typical multi-static SAR system. The AF can be regarded as non-coherent summation of the AF of multiple bistatic systems [33]. The GAF of the multi-satellite fusion system can be expressed as:(14)$$\chi (\vec{r})=\sum_{n=1}^{m}g_{n}(\tau_{d}(\vec{r})M_{n}(f_{d}(\vec{r}))=\sum_{n=1}^{m}P_{n}B_{n}T_{n}\left|\frac{\sin\left(\frac{\pi B_{n}}{c}{\vec{\beta }}_{n}\cdot \vec{r}\right)}{\frac{\pi B_{n}}{c}{\vec{\beta }}_{n}\cdot \vec{r}}\right|\left|\frac{\sin\left(\frac{\pi T_{n}d{\vec{\beta }}_{n}}{\lambda du}\cdot \vec{r}\right)}{\frac{\pi T_{n}d{\vec{\beta }}_{n}}{\lambda du}\cdot \vec{r}}\right|$$

m is the number of satellites participating in the fusion, $B_{n}$ is bandwidth of GNSS signal, $P_{n}$ is power, $T_{n}$ is synthetic aperture time, and ${\vec{\beta }}_{n}$ is the unit vector along the bisecting line of the bistatic angle of the system. It can be seen from the above Formula (14) that the GAF of the multi-satellite fusion system is related to the form of the transmitted signal, the configuration of the transceiver station, and the synthetic aperture time. In [28], the MPSF was defined as:(15)$$\mathrm{MPSF}(\vec{r})=\frac{1}{m}\sum_{n=1}^{m}g_{n}(\tau_{d}(\vec{r})M_{n}(f_{d}(\vec{r}))$$

The MPSF is a noncoherent combination of the individual PSFs, which resolution unit area is the intersection of the single-satellite PSFs. Using the 3dB projection area of the MPSF on the XOY plane as the 2-D resolution unit. Based on the radar parameters in Table 1 and the simulation scene as shown in Figure 2, we can be evaluated using both numerical and analytical methods. The results of MPSF can be approximated as
(16)$$\mathrm{MPSF}=\frac{1}{m}\sum_{n=1}^{m}A_{m}(\varphi ,\gamma ,\alpha )$$

#### 2.3.1. MPSF Numerical Simulation (*m* = 2)

Assume *m* = 2, the quantitative analysis of the MPSF are carried out, assume $A_{1}(180^{\circ },90^{\circ },45^{\circ })+A_{2}(\varphi_{2},\gamma_{2},45^{\circ })$. The optimal MPSF $A_{1}(180^{\circ },90^{\circ },45^{\circ })+A_{2}(100^{\circ },50^{\circ },45^{\circ })$ is shown in Figure 5a, its 2-D resolution area is 743 m^2^, compared with the $A_{1}(180^{\circ },90^{\circ },45^{\circ })$, the area is reduced by 47.2%. The 2-D resolution units of MPSF (*m* = 2) are shown in Figure 6a, Table 2 shows the relevant parameters of the 2-D resolution unit of the optimal MPSF. The value of the highlighted part in Figure 6a reveal that the area of the fused 2-D resolution unit has exceeded 1390 m^2^, indicating that the fusion has not improved the 2-D resolution, but worsened it, meaning that the fusion under this geometric configuration is meaningless. As shown in Figure 7b, compared to the optimal resolution (1390 m^2^) of the single-satellite PSF, about 76.7% 2-D resolutions of MPSF (*m* = 2) had improved in varying degree, and about 23.3% 2-D resolutions were deteriorated.

#### 2.3.2. MPSF Numerical Simulation (*m* ≥ 3)

Assume *m* = 3, the quantitative analysis of the MPSF are carried out, assume $A_{1}(180^{\circ },90^{\circ },45^{\circ })+A_{2}(100^{\circ },50^{\circ },45^{\circ })+A_{3}(\varphi_{3},\gamma_{3},45^{\circ })$, the area of 2-D resolution units are shown in Figure 6b. Table 2 shows the relevant parameters of the 2-D resolution unit of the optimal MPSF (*m* = 3), the optimal MPSF is shown in Figure 5b, its 2-D resolution area is 616 m^2^, compared with the $A_{1}(180^{\circ },90^{\circ },45^{\circ })$, the area is reduced by 55.7%. It shows that the fusion of three-satellite can obtain better resolution performance than the fusion of dual-satellite. Compared with the dual-fusion optimal solution (743 m^2^), only about 30% 2-D resolution of MPSF (*m* = 3) has been improved in varying degrees, and about 70% 2-D resolutions were deteriorated. The results show that the three-satellite fusion has higher requirements for the geometric configuration of the satellites participating in the fusion than the dual-satellite fusion, and the complexity of satellite selection is further improved, but the performance improvement efficiency is reduced.

The MPSF results of the simulation were analyzed (assume *m* ≤ 9). As shown in Figure 5, the MPSF with a well-defined central response at the target position is surrounded by a series of side-lope. With the increase of m, the number of side-lopes increase, and the peak side-lobe ratio (PSLR) decreases. That will cause the image SNR to drop and require more post-processing work. As shown in Figure 7a, the optimal 2-D resolution unit area of the MPSF gradually decreases as m increases, indicating that the resolution improvement as the number of fusion satellites increases. However, the improved efficiency of resolution will be significantly reduced. The simulation results were shown in Figure 7b show that as m increases, compared with the optimal solution of *m*−1, the ratio of combinations with improved resolution decreases, while the ratio of combinations with decreased resolution increases. This means that as m increases, the geometric configuration requirements of the satellite combination with improved resolution will also increase, and the complexity of satellite selection will also increase. In this paper, we considered the following four aspects: resolution improvement efficiency, satellite selection complexity, introduce noise, and computation quantity. In summary that dual-satellite fusion is the cost-effective choice.

## 3. Improved Dual-Satellite Fusion Imaging Method

With the development of multiple navigation systems (GPS, Beidou-2, GLONASS, GALILEO), the average number of visible navigation satellites at the same place on the earth’s surface will rise from 10 to about 40. Therefore, the observation information of GNSS-based Bistatic SAR systems will increase significantly and reduce the revisit period. However, the complexity of satellite selection and computation will increases follow the satellites number increase. According to the Section 2 of the simulation analysis, this section proposes an optimized dual-satellite fusion imaging method, obtaining the optimal 2-D resolution image of the dual-satellite fusion in the scene. To test the performance of this method, the method has been implemented in MATLAB R2018b (MathWorks, Natick, MA, USA) as an interpreted language, and because it is widely used in research and academia. The improved dual-satellite fusion imaging method is divided into two parts. The first part is the satellite selection strategy, the implementation of which is shown in the blue box in Figure 8 and will be explained in detail in Section 3.1; as shown in the red box in Figure 8, the second part is the fusion imaging algorithm, which we will elaborate on in detail in Section 3.2.

### 3.1. Satellite Selection Strategy

For obtaining an optimal dual-satellite fusion resolution image, it is necessary to select the best combination from a large number of visible satellites for fusion imaging. To reduce computation and obtain better imaging resolution, a satellite selection method based on a greedy algorithm is proposed. The proposed method linearly approximates the best 2-D resolution of the dual-satellite fusion by successive greedy selection of two satellites in an iterative way. The specific steps are shown in Figure 8, Algorithm 1 gives the specific implementation of the pseudo code.

• Step 1: Select a subset of navigation satellites for imaging

Before imaging, using the navigation satellite ephemeris data to roughly determine the visible satellites distribution at the receiver’s location, and build the geometric configuration model of GNSS based Bistatic SAR system based on the location information of the target area. Select a subset of navigation satellites for imaging based on the geometric model. In consideration of two important factors related to image quality, the first is the imaging resolution; the second is the signal-to-noise ratio (SNR) of the image, which is related to the received signal power. Therefore, the image quality is related to the geometric configuration of the GNSS-based bistatic SAR. The resolution can be determined by the Formula (7) in the range and azimuth direction. Combined with the power analysis of the navigation signal backscatter [34], the subset of navigation satellites for imaging can be constrained by constraining its azimuth angle $120^{\circ }\leq \varphi \leq 240^{\circ }$ and elevation angle $10^{\circ }\leq \alpha \leq 70^{\circ }$.

• Step 2: select the reference satellite

Using a greedy selection strategy, based on the geometric configuration information provided by the scenario, the PSF of each satellite is estimated by (12), and the reference satellite corresponding to the optimal PSF is selected.

• Step 3: auxiliary satellite selection

The satellite that can be fused with the reference satellite to obtain the best imaging resolution is selected as the auxiliary satellite. Numerical analysis shows that the minimum area of 2-D resolution is to be observed when the two PSF orientations η are nearly orthogonal. Therefore, the angle $\psi =\left|\eta_{R}-\eta_{m}\right|$ between the PSF orientation $\eta_{R}$ of the reference satellite and the PSF orientation $\eta_{m}$ of the subset of navigation satellites can be calculated, and when $\eta_{R}$ and $\eta_{m}$ are orthogonal to each other ($\psi =90^{\circ }$), the system achieved the optimal 2-D resolution; when $\eta_{R}$ and $\eta_{m}$ parallel ($\psi =0^{\circ }$ or $\psi =180^{\circ }$), the 2-D resolution unit of the reference satellite overlaps the 2-D resolution unit of the fusion satellite, the system achieved the worst resolution. Therefore, find the auxiliary satellite in the subset of navigation satellites by an iterative search method, when ψ is closest to 90°, the corresponding satellite is selected as the auxiliary satellite.

The pseudo-codes of satellite selection strategy are listed as follows:
**Algorithm 1: Satellite Selection Strategy** Load the raw data; Calculate receiver position and visible satellites *n*; **for visible satellites *n***  Calculate the azimuth angle φ and elevation angle α;  Perform threshold comparison for φ and α;  Obtain a subset of imaging satellites *m*; **end**  Initiate the processing parameters;   **for imaging satellites *m***    Calculate the area $S_{m}$ and direction of PSF $\eta_{m}$;    Search for minimum area $S_{m}$;    Search for the best angle ψ;    Obtain reference satellite and auxiliary satellite;  **end**

### 3.2. Imaging Algorithm

For better imaging results, we chose the traditional time-domain imaging algorithm, the block diagram of which is shown in Figure 8, Algorithm 2 gives the specific implementation of the pseudo code. The proposed algorithm is an improvement of the (Back projection) BP algorithm, a specific process can be divided into four parts.

1. Signal Synchronization

GNSS-based Bistatic SAR is mainly faced with three synchronization problems: (1) space synchronization; (2) time synchronization; (3) frequency and phase synchronization. The article [35] gives a detailed description of the synchronization problem, and we will not repeat it here.

2. Range compression and range migration correction

It is well known that the echo channel of the GNSS-based Bistatic SAR system can receive echo signals from multiple satellites. When processing the echo signals, the echo signals from different satellites are superimposed on each other. The pseudo-random code (PRN) in the navigation signal has particularly excellent auto-correlation and cross-correlation characteristics (taking GPS C/A code as an example, the ratio of the auto-correlation peak to the maximum cross-correlation peak is about 24 db). Therefore, the corresponding matched filters can be constructed through the PRN codes of different satellites to achieve range compression without aliasing. First, the carrier phase and code phase information of the satellite’s direct signal reconstruct the direct signal, and perform synchronization processing on the reconstruct direct signal and the echo signal, respectively perform range FFT on the reference signal and the echo signal, and convert the signal to time domain and multiplied by the complex conjugate of the reference signal. The result uses Inverse Fast Fourier Transform (IFFT) to complete the range compression of GNSS-bistatic SAR, and the signal is expressed in the following form.
(17)$$F(t,u)=R_{x}\left[u-\frac{R(t)}{c}\right]\omega \left(u-u_{ta}\right)exp\left[-j2\pi f_{c}\frac{R(t)}{c}\right]$$
where $R_{x}\left[\cdot \right]$ denotes the cross-correlation function between the received and reference signals in the range-time dimension. $\omega \left(u-u_{ta}\right)$ is the envelope of the received signal in the azimuth-time dimension, and $f_{c}$ is the carrier frequency of the transmitted signal, and $R(t)=R_{T}(t)+R_{R}(t)-R_{B}(t)$. The instantaneous navigation satellite-to-target range is denoted by $R_{T}(t)$, $R_{R}(t)$ is the instantaneous receiver-to-target range, and $R_{B}(t)$ is the instantaneous satellite-to-receiver range. Due to the different equivalent squint angle caused by the linear component of the residual satellite range, the Range Cell Migration (RCM) of the data in the same range cell has a different slope in the azimuth time domain. Although in GNSS-based Bistatic SAR, the transmitted signal is no longer a chirp signal, the signal still has chirp characteristics, so we can use the chirp scaling algorithm to process RCM Correction (RCMC). The filter of RCMC can be expressed as:(18)$$H_{r}=exp(j2\pi f_{r}\frac{R(t)}{c})$$
where $f_{r}$ is the range frequency, transform the time-domain range compression result F(t,u) into the range-frequency domain $F(f_{r},t)$ through the Fast Fourier Transform (FFT) of the range, multiply it with the range migration correction filter, and then perform Inverse Fast Fourier Transform (IFFT) to complete the range migration correction. The signal is converted back to the time domain, it can be expressed as
(19)$$F_{c}(t,u)=IFFT\left[F(f_{r},t)\cdot H_{r}\right]$$

3. Azimuth compression and compensate the Doppler phase

Unlike the RD algorithm, the azimuth compression of the BP algorithm is completed in the time domain, and the specific implementation method is the coherent accumulation of the range compression signal and compensate the Doppler phase in the synthetic aperture time. The azimuth compressed signal can be expressed as
(20)$$g(t,u)=\sum_{i=1}^{t}\left(F_{c}(t,u)exp(-j2\pi \frac{R(t)}{\lambda })\right)\left(i\cdot n\right)$$
where *n* is the azimuth sampling point.

4. Fusion imaging results

Using the proposed imaging algorithm, the imaging results of the reference satellite and the auxiliary satellite are obtained respectively. After image registration and amplitude equalization, the final dual-satellite fusion image can be obtained by the noncoherent addition.

The pseudo-codes of fusion imaging algorithm are listed as follows:
**Algorithm 2: Fusion Imaging Algorithm** Load the raw data; Initiate the processing parameters; **for reference satellite and auxiliary satellite**  Reconstructe reference signal  Range compression;  Image buffer set zero;  **for every pixel in the image**   **for every PRI in an aperture**    Calculate the slant range;    Calculate the echo location;    Compensate the Doppler phase;    Azimuth compression;   **end**  **end**  Fusion imaging results; **end**

## 4. Simulation and Discussion

In this section, we demonstrate the validity and feasibility of the proposed method. Potentially, the proposed method can be applied to all constellations of the GNSS system. The 24 h real GPS-L1 orbit data were used for point targets imaging simulation experiment. The precision ephemeris is provided by the International GNSS Service (IGS), which offers three types of precision ephemeris: IGS Final Precision Ephemeris (IGS Final), the IGS Rapid Precision Ephemeris (IGS Rapid), and IGS Ultra-Rapid Precision Ephemeris (IGS Ultra-Rapid). In this paper, we chose the IGS Ultra-Rapid to develop the simulation because it has good real-time performance, with an orbital position error of less than 10 cm and a satellite clock error of less than 5 ns. The separation time interval of IGS Ultra-Rapid data is 15 min, it is too long for the sampling interval 1 ms of GNSS-based Bistatic SAR system, Lagrange interpolation was used to complete the data fitting. Table 3 lists the specific parameters, and the distribution of the simulation scene as show in Figure 9.

In Figure 10, the corresponding value of the diamond represented the area of the optimal resolution unit obtained by the proposed method, and the corresponding value of the circle represented the area of the global optimal resolution unit obtained by the traversal fusion method. In 24 data sets, the optimal resolutions generated by dual-satellite fusion were significantly better than the optimal resolutions generated by single-satellite. Compared to global optimal dual-satellite fusion resolution, our proposed method generated 15 times the global optimal resolution and 8 times the suboptimal resolution, which showed that the validity and feasibility of the method.

To show the performance of the proposed method, the area of the optimal resolution units generated by the proposed method were compared with the area of the optimal resolution units of single-satellite. A set of data was selected from 24 data sets for detailed analysis, the data collection time was at 18:00. Using the proposed satellite selection strategy, in Figure 11, there were 4 navigation satellites in the orbit that can be used for imaging, the satellite PRN21 was selected as the reference satellite, and PRN10 was selected as the auxiliary satellite. The single-satellite imaging results of PRN21 and PRN 10 are demonstrated in Figure 12a,b, the dual-satellite fusion imaging result with our proposed algorithm are shown in Figure 12c, it can be seen that significantly the resolution units of dual-satellite fusion imaging results are smaller than single-satellite.

The 2-D resolution was characterized by the area of an approximate ellipse in the 3 dB plane. To better visualize the performance gains resulting from the proposed fusion method, the imaging results have projected onto the 3 dB plane. Figure 13c shows the 3 dB image response obtained the fusion of PRN21 and PRN 10, which area is 853 m^2^, compared with the PRN 21 (1579 m^2^) in Figure 13a, the area is reduced by 45%. The comparative analysis results fully prove that the proposed method has the excellent resolution improvement performance.

## 5. Experimental Verification and Analysis

To demonstrate the validity and feasibility of our proposed fusion imaging formation method, real scene experiments were carried out. To obtain better resolution performance, according to the known accurate ephemeris data, accurate satellite orbit information was obtained, and the following experiment was designed. Figure 14 shows the relevant configuration of the experiment, and Table 4 lists the detailed parameters. The experimental hardware is designed by Beihang University and contains four navigation signal receiving channels. It is an X86 architecture data acquisition and processing platform that supports GPS L1/L5 and BeiDou B1/B3 data acquisition and processing. The right-handed circularly polarized (RHCP) omnidirectional antenna is GPS-702-GG (Novatel, Calgary, AB, CAN) to receive the direct signal of GNSS, which is used to realize system positioning and obtain accurate carrier phase and code phase information to provide reference information for signal synchronization. The echo channel uses a high-gain left-handed circular polarized (LHCP) antenna receives the echo signal from the target area. The LHCP antenna is a 4-array microstrip antenna designed by Beihang University, the specific parameters are shown in Table 4. The data collection location is the roof of Block E of the new main building of Beihang University. The target area is mainly the gymnasium of Beihang University, which is an all-metal roof with a large RCS, that is beneficial for imaging.

The experiment was conducted at 20:00 on 20 August 2020. Based on our proposed satellite selection strategy, there are 3 navigation satellites in this area that can be used for imaging (PRN12, PRN15, PRN24), GPS satellite PRN15 was selected as the reference satellite, and PRN24 was selected as the auxiliary satellite. The imaging results of the navigation satellites obtained through the improved BP algorithm are shown in Figure 15a–c. Different images of the gymnasium were obtained under different geometric configurations. The gymnasium is a 100 × 100 m building, subject to the resolution ability of GPS L1 signals and geometric configuration, the obtained image can only reflect that the stadium is a point target in the range direction, and it can be divided into discrete point targets in the azimuth direction. As shown in Figure 15d, the experimental results are consistent with the theoretical calculation.

Figure 16 shows the imaging results of dual-satellite fusion, which were generated by the non-coherent addition of the imaging results of the single-satellite. When compared with the area of the minimum resolution unit of Figure 15a, which were reduced about 70.1%, 64.2% and 41.4% in Figure 16a–c, the improvement of the resolution brought dual-satellite fusion was obvious. Compared to the resolution of Figure 16b,c, the Figure 16a was improved most obviously, which was generated by our proposed method. The specific parameters are shown in Table 5, which proves the feasibility and effectiveness of this method. The obtained image is compared with the optical image of the target area. It can be seen from Figure 15d that the stadium can only reflect discrete point targets distributed along with the azimuth direction, due to limited by the range resolution. As shown in Figure 16d, it can be seen that the discrete surface targets revealed in the radar image of the stadium more accurately show the feature information of the stadium, due to the increase of resolution.

In the experiment, *m* = 3, the total calculation of the proposed method can be approximated as $\left[N_{a}N_{r}\log_{2}N_{r}+N_{a}N_{r}+N_{kel}N_{x}N_{y}N_{a}\right]*2$, the total computational load for the iterative fusion method can be represented by $\left[N_{a}N_{r}\log_{2}N_{r}+N_{a}N_{r}+N_{kel}N_{x}N_{y}N_{a}\right]*3$, where $N_{a}$ and $N_{r}$ are the number of range and azimuth samples, respectively. $N_{x}$ and $N_{y}$ are the numbers of samples of the imaging scene, and $N_{kel}$ is the length of the interpolation kernel. The experiment is carried out in MATLAB 2018b (MathWorks, Natick, MA, USA) on a computer with AMD R7-3800X 3.9 GHz processors and 64 GB RAM. In the experiment, the time consumed using the proposed algorithm was 1970 s, while that of the iterative fusion method was 2955 s. The experiment demonstrated the feasibility of the proposed method by showing that it achieves good imaging resolution while reducing the computational effort.

## 6. Conclusions

In this paper, we focused on the influence of geometric configuration on the fusion resolution of GNSS-based bistatic SAR system. The theoretical analysis and numerical simulation of MPSF are carried out. The influence of geometric configuration and the fusion satellite number on the MPSF are evaluated in combination with point target simulation. It is concluded that dual-satellite fusion is a cost-effective combination. An improved dual-satellite fusion imaging method is proposed. It consists of two parts, a satellite selection strategy and an improved BP algorithm. As demonstrated by simulation and experiment results, the proposed method performed well in terms of both imaging resolution and efficiency. It is worth noting that the method is more effective at long integration times and in airborne mode, where the azimuthal resolution is about 1 m and the area of the fused resolution unit can be reduced to about 1 m^2^. The next step will be a study of airborne mode or multi-system signal fusion with long synthetic aperture time to assess the full potential and challenges of the proposed approach.

In terms of the proposed method itself, we found three application difficulties, which can be improved in the future work. 

(1)Multi-satellite fusion has resulted in improved resolution, however, it has introduced a series of side-lope. This will result in a degradation of image quality, however, we plan to use the clean algorithm [29] to solve this problem.(2)The proposed satellite selection strategy is based on the greedy algorithm, and it is easy to fall into local optimum. Therefore, we can optimize the satellite selection strategy to improve the imaging resolution in the follow-up work.(3)The fusion imaging algorithm is realized by an improved BP algorithm, however, it has a large amount of calculation, and we can use an frequency domain imaging method to improve the computational efficiency.

## Figures and Tables

**Figure 1 sensors-20-07119-f001:**
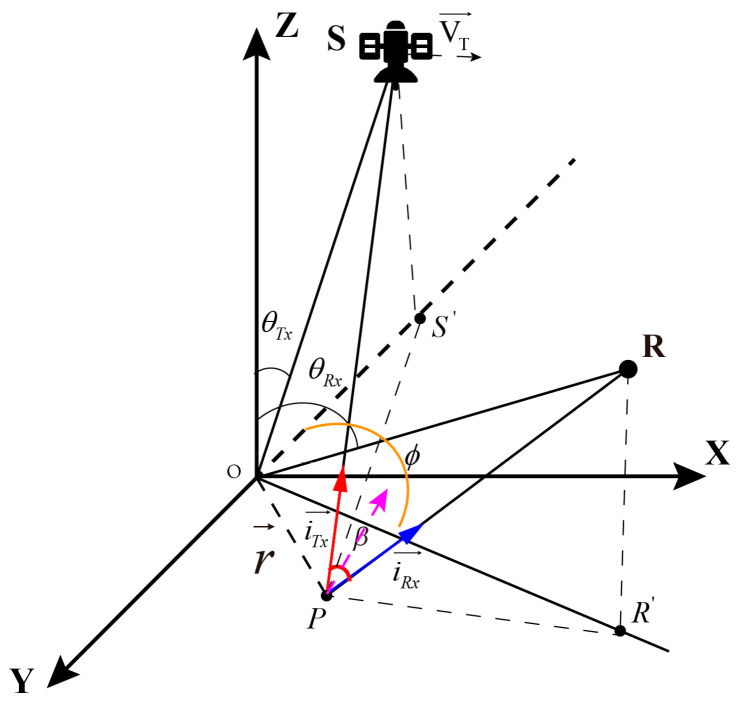
GNSS-based Bistatic SAR geometry—a ground-based stationary receiver.

**Figure 2 sensors-20-07119-f002:**
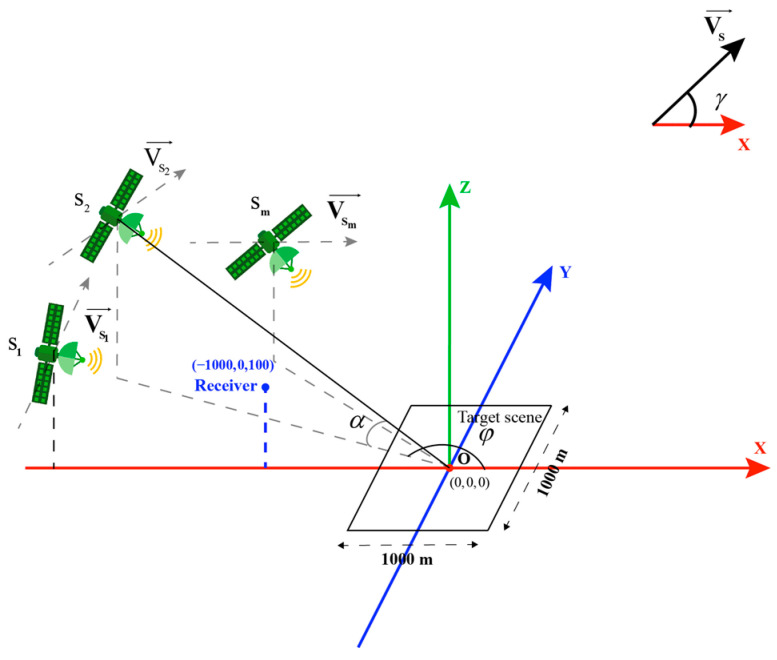
Simulation scene geometry configuration diagram.

**Figure 3 sensors-20-07119-f003:**
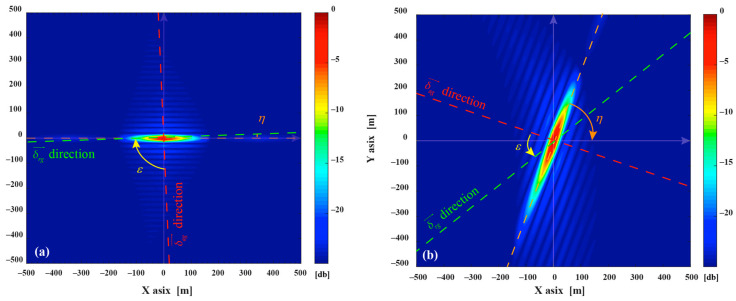
Simulated single-satellite Point Spread Function (PSFs), (**a**) is the PSF of $A_{1}(180^{\circ },90^{\circ },45^{\circ })$, (**b**) is the PSF of $A_{1}(150^{\circ },60^{\circ },45^{\circ })$.

**Figure 4 sensors-20-07119-f004:**
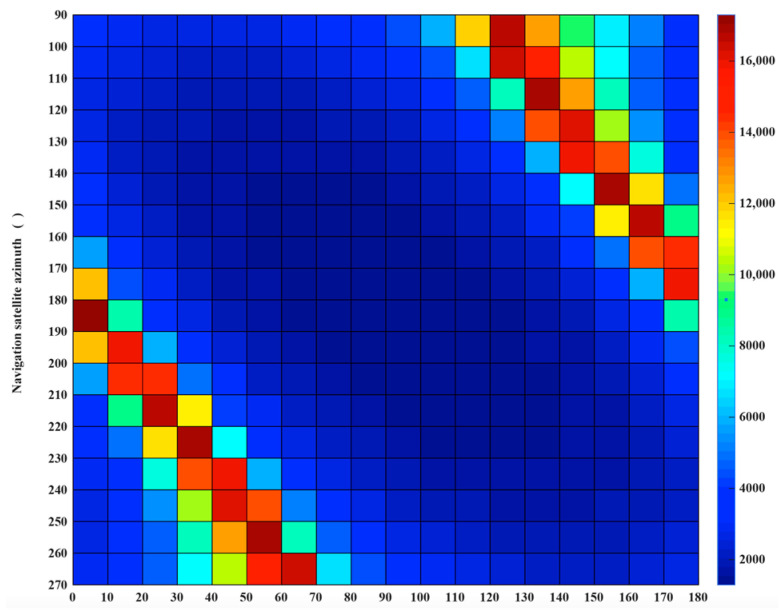
Two-dimensional (2-D) resolution unit area distribution of $A_{1}(\varphi ,\gamma ,45^{\circ })$.

**Figure 5 sensors-20-07119-f005:**
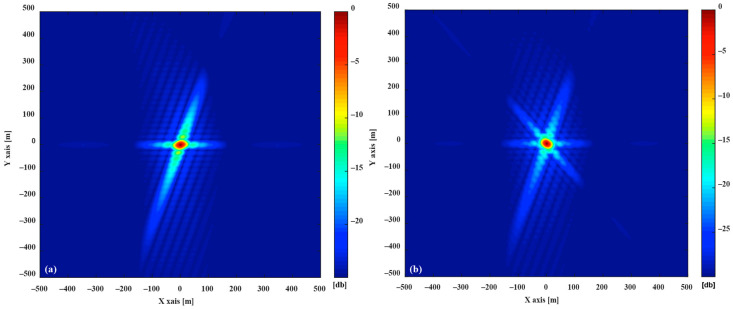
Simulated multi-satellite fusion MPSF: (**a**) $A_{1}(180^{\circ },90^{\circ },45^{\circ })+A_{2}(100^{\circ },50^{\circ },45^{\circ })$, (**b**) $A_{1}(180^{\circ },90^{\circ },45^{\circ })+A_{2}(100^{\circ },50^{\circ },45^{\circ })+A_{3}(120^{\circ },30^{\circ },45^{\circ })$.

**Figure 6 sensors-20-07119-f006:**
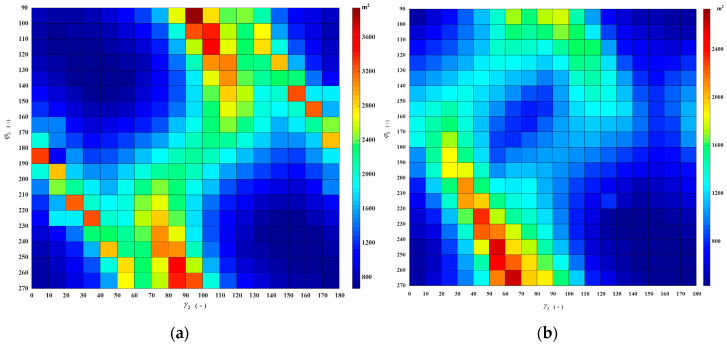
Two-dimensional (2-D) resolution unit area distribution of the Multi-Satellite Fusion Point Spread Function (MPSF): (**a**) $A_{1}(180^{\circ },90^{\circ },45^{\circ })+A_{2}(\varphi_{2},\gamma_{2},45^{\circ })$, (**b**) $A_{1}(180^{\circ },90^{\circ },45^{\circ })+A_{2}(100^{\circ },50^{\circ },45^{\circ })+A_{3}(\varphi_{3},\gamma_{3},45^{\circ })$.

**Figure 7 sensors-20-07119-f007:**
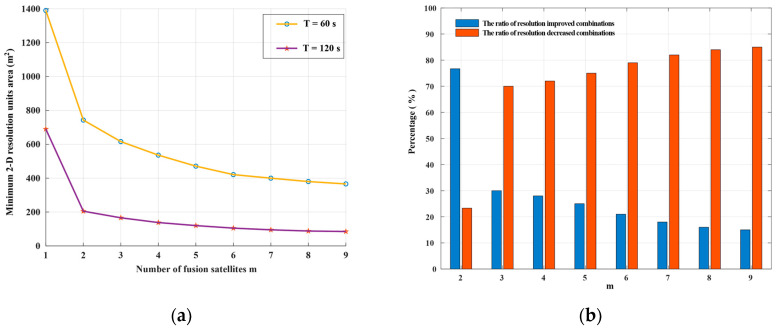
The optimal 2-D resolution of MPSF (*m* ≤ 9), (**a**) is the variation of the minimum 2-D resolution units area with m (*T* = 60 s and *T* = 120 s), (**b**) is the ratio of resolution increase and decrease compared to the optimal resolution of *m*−1 (*T* = 60 s).

**Figure 8 sensors-20-07119-f008:**
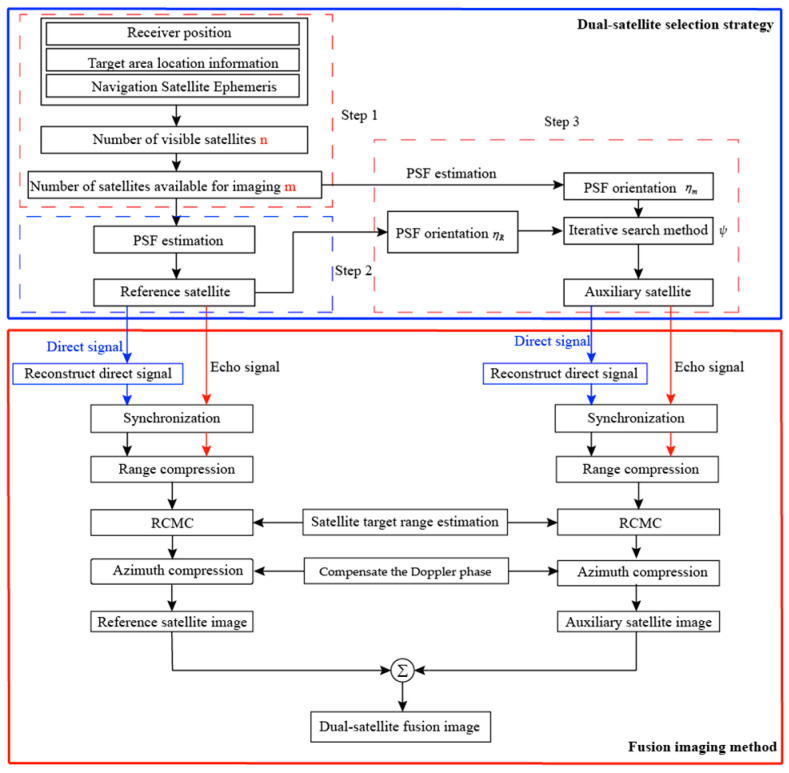
Flow chart of the improved dual-satellite fusion imaging method.

**Figure 9 sensors-20-07119-f009:**
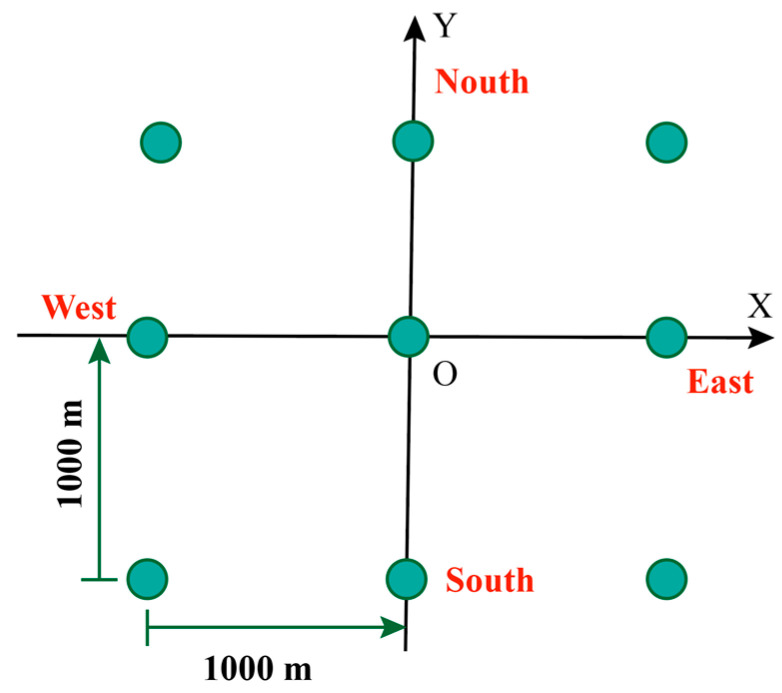
Simulation scene.

**Figure 10 sensors-20-07119-f010:**
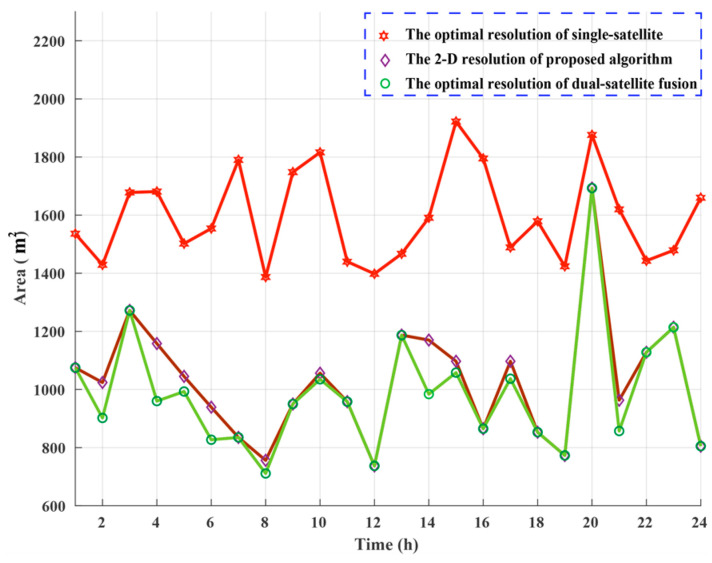
Distribution of the area of the resolution units in 24 h.

**Figure 11 sensors-20-07119-f011:**
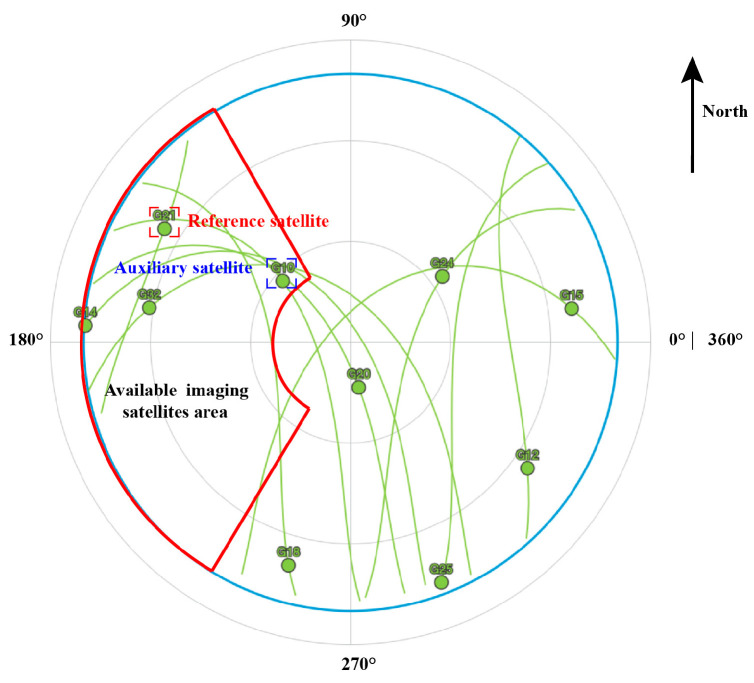
Sky map at 18:00.

**Figure 12 sensors-20-07119-f012:**
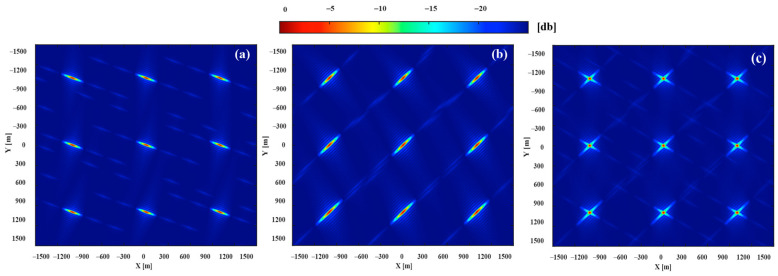
The imaging results: (**a**) is the imaging results of PRN21, (**b**) is the imaging results of PRN10, and (**c**) is fusion imaging results of PRN21 and PRN10.

**Figure 13 sensors-20-07119-f013:**

The 2-D resolution units: (**a**) is the 2-D resolution units of PRN21, (**b**) is the 2-D resolution units of PRN10, and (**c**) is the fusion 2-D resolution units of PRN21 and PRN10.

**Figure 14 sensors-20-07119-f014:**
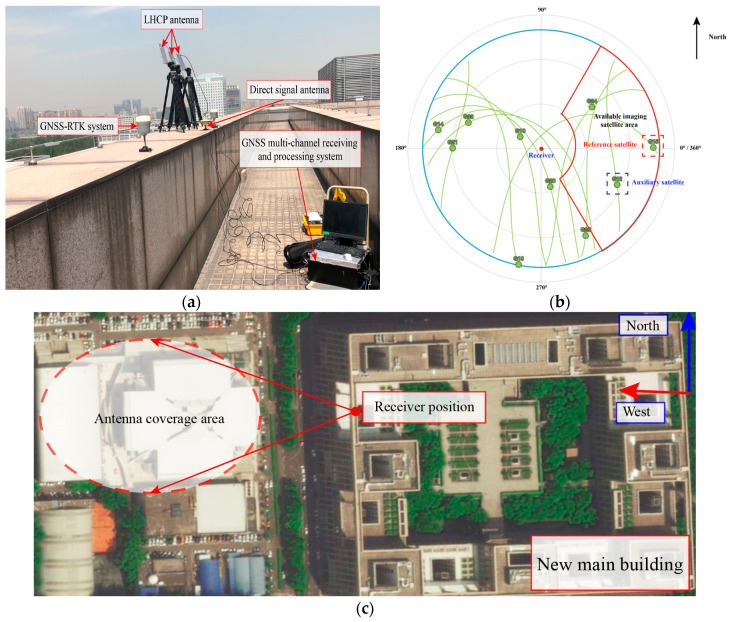
Experiment specific configuration: (**a**) experimental equipment hardware, (**b**) sky map at the time of collection, (**c**) optical image of experimental area.

**Figure 15 sensors-20-07119-f015:**
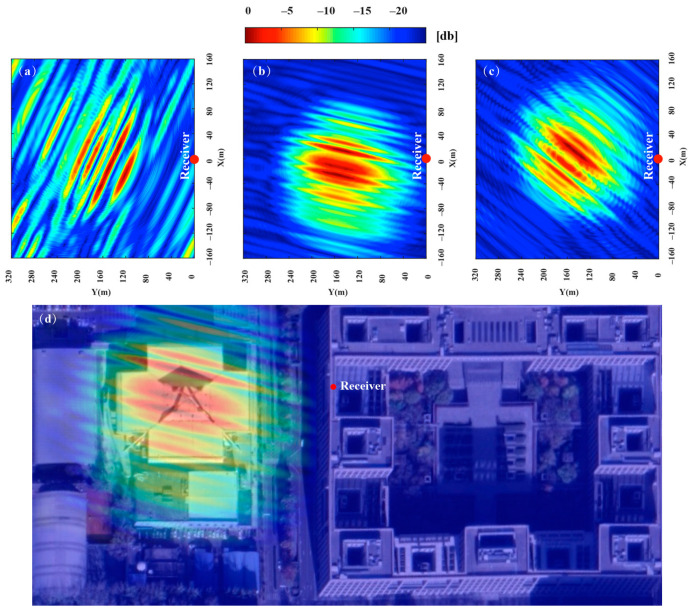
The imaging results: (**a**) is the imaging results of PRN15, (**b**) is the imaging results of PRN12, (**c**) is the imaging results of PRN 24, (**d**) is the imaging results of PRN12 and optical image (Google Map) matching.

**Figure 16 sensors-20-07119-f016:**
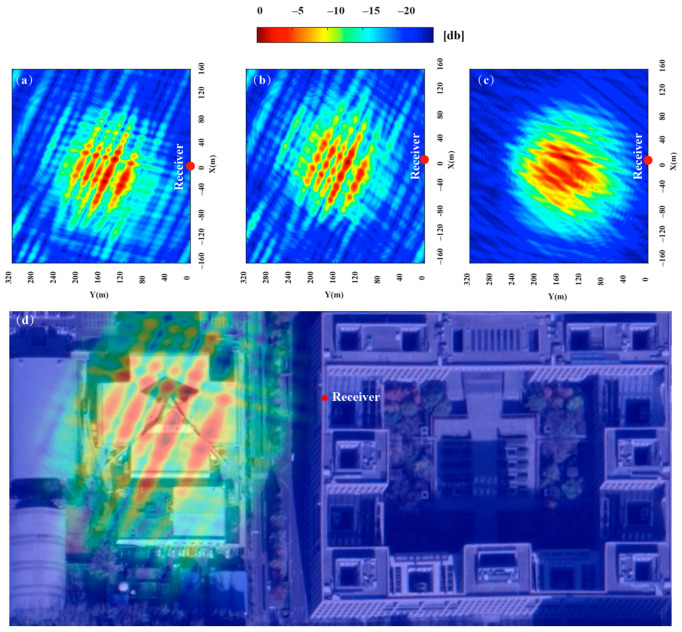
Dual-satellite fusion image: (**a**) fusion image of PRN15 and PRN12, (**b**) fusion image of PRN15 and PRN24, (**c**) fusion image of PRN12 and PRN24, (**d**) dual-satellite image and optical image (Google Map) matching.

**Table 1 sensors-20-07119-t001:** Simulation parameters.

Parameter	Value
Satellite operating speed	3000 m/s
$S_{m}$ Initial azimuth	($\varphi_{m}:90^{\circ }\sim 270^{\circ }$)
Angle sampling interval	$10^{\circ }$
$S_{m}$ velocity direction	($\gamma_{m}$$:0^{\circ }\sim 180^{\circ }$)
α	$45^{\circ }$
Slope distance of target center	20,000 km
Synthetic aperture time	60 s
Signal band width (GPS C/A)	2.046 MHz
Carrier frequency	1575.42 MHz
Receiver and Point target position	(−1000,0,100) (0,0,0) m

**Table 2 sensors-20-07119-t002:** Simulated MPSF 2-D resolution unit area (*m* = 2 and *m* = 3).

Sat	$\left|\delta_{rg}\right|$	$\left|\delta_{ag}\right|$	φ	γ	α	S
Ref	m	m	${}^{\circ }$	${}^{\circ }$	${}^{\circ }$	$m^{2}$
$A_{1}$	131.53	21.1	180	90	45	1390
$A_{2}$	138.24	24.03	100/230	50/140	45	2379
$A_{1}+A_{2}$	~	~	~	~	45	735
$A_{3}$	152.65	25.7	240/120	150/30	45	1974
$A_{1}+A_{2}+A_{3}$	~	~	~	~	45	616

**Table 3 sensors-20-07119-t003:** Simulation parameters.

Parameter	Value
Data collection time	17 September 2020 1:00~24:00
sampling interval	1 h
Synthetic aperture time	60 s
Signal band width (GPS C/A)	2.046 MHz
Sampling rate	62 MHz
PRF	100 Hz
Receiver position	(−1000,0,100) m

**Table 4 sensors-20-07119-t004:** Parameters used in the experiment.

Parameter	Value
GPS satellite	PRN15, PRN12, PRN24
Satellite starting angle (Pitch, Azimuth)	($14.25^{\circ }$,$2.1^{\circ }$)($33.18^{\circ }$,$335.24^{\circ }$)($45.75^{\circ }$,$38.89^{\circ }$)
Select signal	L1 (1575.42 MHz)
Bandwidth	2.046 MHz
RHCP antenna gain	3 dBi
LHCP antenna beamwidth	$\pm 19^{\circ }$
LHCP antenna gain	13 dBi
RHCP /LHCP antenna working Frequency Band	GPS L1/BeiDou B1
System sampling rate	62 MHz
Quantization bit	14 bit
Synthetic aperture time	60 s

**Table 5 sensors-20-07119-t005:** Experimental parameters.

Sat	$\left|\delta_{rg}\right|$	$\left|\delta_{ag}\right|$	ψ	ΔS (Experimental Measurements)
Ref.	m	m	(∘)	$m^{2}$
PRN15	135.25	25.710	~	2667.9
PRN12	158.99	22.52	~	2939.8
PRN24	181.01	24.80	~	3347.3
PRN15∩PRN12	~	~	81	799
PRN15∩PRN24	~	~	138	953.56
PRN12∩PRN24	~	~	34	1563.24

ΔS are the experimental measurement that extracts the highlighted partial resolution units from the real image and calculates the area of its 3dB plane approximating the ellipse by Monte Carlo method.

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
