# Peer review of "Improved GNSS-Based Bistatic SAR Using Multi-Satellites Fusion: Analysis and Experimental Demonstration"

_sensors, 2020, doi:10.3390/s20247119_

Round 1

Reviewer 1 Report

Dear authors,

Your manuscript provides a sound overview of your proposed methodology, and the verification results show a good match with what you expected to find from the introduction. 

Some remarks from my side:

  • The integration of GNSS-R and SAR could not be clearly found in the manuscript. You should include another section to highlight that and how the integration can improve remote sensing based application results.
  • You should also include a section on the integration of other GNSS providers apart from BeiDou, such as GLONASS, Galileo, etc. What will be the problems to solve when combining GNSS data from different providers.
  • Another point is your finding that a combination of the signal from two satellites can improve more significantly the overall results that in the case of a combination of multiple satellite. I agree that this can happen, however the consideration of a couple of satellites can introduce errors that  can be reduced/mitigated by the consideration of multiple satellites. Make more clear in your analysis why the use of only two satellites is preferable apart from the more difficult and time-consuming data processing.
  • You should include more information about the difficulties/obstacles you found in applying your methodology.
  • As concluding remarks, you should analyse further ways to overcome the obstacles/problems and provide a better overview of the next planned steps of research to improve your study and possibly your results.

Author Response

Dear reviewer

We are truly grateful to the comments and suggestions from your review. Based on these comments and suggestions, we have made careful modifications on the manuscript. 

Please refer to the attachment for specific modifications

Reviewer 2 Report

The manuscript presents an important study concerning SAR surveys improved by applying satellite imaging fusion in order to achieve better resolution and increase the overall efficiency. Both the article's merit and its scientific soundness reveals a high level of the research. What is more, the investigations presented were funded by a noble institution. Also, the material is significant, as well as its practical usefulness is noticeable. Nevertheless, the text contains some shortcomings which should absolutely be considered before the text is published. 

Lines 63-64: Increasing the resolution seams here to be a problem that needs to be solved. Unfortunate sentence - I suggest to rephrase it.

Lines 85-88: These sentences can be misleading. Either the increase of satellite images taken into account provide us with better results or the accuracy drops down. I understand what the authors meant, but in my opinion, the caption should be rephrased.

Lines 109-126: This section presents formulas quoted from different publications. This is the state-of-the-art, and the text needs some explanation about the usefulness of the formulas in further considerations.

- The text lacks sufficient, fair information about the software used (mainly for data modelling and for its visualizing). From Line 459, one case concludes that the software used was own elaboration. If so, please provide more information in the text.

- Figure 8 refers to section 3.1 and hence, should appear in that section.

- Is the flowchart presented in Figure 8 an own elaboration? If so, please provide readers with relevant information about it.

- Lines 379-391: One can assume, that the equipment presented was locally designed; does it refer to the overall test station or rather to the electronic equipment? On the picture, a GNSS-antenna can easily be spotted. Is it also a self-design? If no, please provide readers with more information (the instrumental part of the study seems to be crucial here).

Section 6 'Conclusions' begins with quite embroiled sentences. What is more, it repeats what has already been written previously. I would suggest rephrasing the section so that it would be more understandable, providing fair consideration what comes out from the preceding text.  

I would be pleased to have a chance to read the improved text in the second reviewing cycle.

Author Response

Dear reviewer

We are truly grateful to the comments and suggestions from your review. Based on these comments and suggestions, we have made careful modifications on the manuscript.

Please refer to the attachment for specific modifications

Best regards,
Shiyu Wu

Reviewer 3 Report

Remarks:

110: … the ambiguity function (AF) can be regarded as the correlation coefficient of the echo of two points that can be spatially distinguished. It is not clear statement.

Define all arguments OA, OB, t, u in the expression (1).

119: It is written: B is signal bandwidth; P is power. Write: B is the signal bandwidth; P is the power.

130: It is written: Figure 1 shows the geometric configuration where navigation satellite S was located on the XOY plane. It is not a correct statement. The satellite is in the coordinate system OXYZ.

199: It is written: P is power, n T is synthetic-aperture time. Write P is the power, Tn is the synthetic-aperture time. English has to be improved.

recommended ref:

Lazarov, V. C. Chen, T. Kostadinov, J.P. Morgado. Bistatic SAR System with GPS Transmitter, in Proceedings of IEEE RADAR Conference 2013, Ottawa, Canada, 29 April -3 May, 2013, pp. 1-6. DOI: 10.1109/RADAR.2013.6586133, https://ieeexplore.ieee.org/document/6586133

Author Response

(The authors gave the same response as above.)

Round 2

Reviewer 2 Report

The improved manuscript contains necessary explanations and completions recommended by the reviewer. The text is of high quality and presents valuable studies, analysis and demonstrations. Regarding that, I appreciate the authors' input in the final version of the article and I recommend to accept the text for publication. Herewith, I would like to thank the author's for considering the reviewer's advises and congratulate them their interesting test.